# Microbial Community Composition of Explosive-Contaminated Soils: A Metataxonomic Analysis

**DOI:** 10.3390/microorganisms13020453

**Published:** 2025-02-19

**Authors:** Francisco J. Flores, Esteban Mena, Silvana Granda, Jéssica Duchicela

**Affiliations:** 1Departamento de Ciencias de la Vida y la Agricultura, Universidad de las Fuerzas Armadas-ESPE, Sangolquí 171103, Ecuador; ejmena3@espe.edu.ec (E.M.); slgranda@espe.edu.ec (S.G.); 2Centro de Investigación de Alimentos, CIAL, Facultad de Ciencias de la Ingeniería e Industrias, Universidad UTE, Quito 170527, Ecuador

**Keywords:** bioremediation, soil microbial diversity, natural attenuation, environmental health, sustainability, biodiversity

## Abstract

Munition disposal practices have significant effects on microbial composition and overall soil health. Explosive soil contamination can disrupt microbial communities, leading to microbial abundance and richness changes. This study investigates the microbial diversity of soils and roots from sites with a history of ammunition disposal, aiming to identify organisms that may play a role in bioremediation. Soil and root samples were collected from two types of ammunition disposal (through open burning and open detonation) and unpolluted sites in Machachi, Ecuador, over two years (2022 and 2023). High-throughput sequencing of the 16S rRNA gene (for bacteria) and the ITS region (for fungi and plants) was conducted to obtain taxonomic profiles. There were significant variations in the composition of bacteria, fungi, and plant communities between polluted and unpolluted sites. Bacterial genera such as *Pseudarthrobacter*, *Pseudomonas*, and *Rhizobium* were more abundant in roots, while *Candidatus Udaeobacter* dominated unpolluted soils. Fungal classes Dothideomycetes and Sordariomycetes were prevalent across most samples, while Leotiomycetes and Agaricomycetes were also highly abundant in unpolluted samples. Plant-associated reads showed a higher abundance of *Poa* and *Trifolium* in root samples, particularly at contaminated sites, and *Alchemilla*, *Vaccinium*, and *Hypericum* were abundant in unpolluted sites. Alpha diversity analysis indicated that bacterial diversity was significantly higher in unpolluted root and soil samples, whereas fungal diversity was not significantly different among sites. Redundancy analysis of beta diversity showed that site, year, and sample type significantly influenced microbial community structure, with the site being the most influential factor. Differentially abundant microbial taxa, including bacteria such as *Pseudarthrobacter* and fungi such as *Paraleptosphaeria* and *Talaromyces*, may contribute to natural attenuation processes in explosive-contaminated soils. This research highlights the potential of certain microbial taxa to restore environments contaminated by explosives.

## 1. Introduction

Explosives are widely used in military and mining operations; however, their utilization, particularly disposal, can lead to soil contamination, which poses a significant threat to the environment and human health [1]. Natural attenuation, or the ability of natural processes to reduce the concentration of contaminants in soil, is a promising approach for remediation of explosive-contaminated soils. Microbes, including bacteria and fungi, play a critical role in natural attenuation by enhancing the degradation or transformation of contaminants into less harmful compounds [2]. Notwithstanding, the microbial diversity and community structure of soils exposed to explosives and their potential role in natural attenuation remain poorly understood.

The most commonly used explosive organic compounds are 2,4,6-trinitrotoluene (TNT), hexahydro-1,3,5-trinitro1,3,5-triazine (RDX), and octahydro-1,3,5,7-tetranitro-1,3,5,7-tetrazocine (HMX). These compounds are susceptible to degradation by *Pseudomonas*, *Bacillus*, *Klebsiella*, *Arthrobacter*, and *Acinetobacter* bacteria, some of them related to *Arabidopsis* and *Populus* roots [3,4,5,6,7]. *Arabidopsis* plants transformed with the AfSSB gene from *Acidithiobacillus ferrooxidans* showed improved tolerance to TNT and cobalt, enhancing phytoremediation in contaminated soils [8]. Some fungi within the phyla Ascomycota and Basidiomycota show RDX and TNT degradation capabilities, whereas *Aspergillus*, *Fusarium*, and *Mortierella* have the potential for natural attenuation [9,10]. Recent studies reported that exposure to munition compounds significantly reduces soil microbial diversity and richness, as indicated by declines in diversity indices like the Shannon index [11]. TNT decreases soil respiratory and enzymatic activity, favoring resilient taxa like *Pseudomonas aeruginosa* and *Sphingomonadaceae* while reducing *Acidobacteria* and *Gemmatimonadetes*. RDX reduces soil biomass and acid phosphatase activity, increasing *Actinobacteria* and *Bacillus*, whereas HMX enhances invertase and catalase activities, promoting *Gammaproteobacteria* and *Acidobacteria* [12]. Mixed exposures further inhibit microbial respiration and enzymatic activity, with *Proteobacteria* and *Burkholderiaceae* becoming dominant. These compounds disrupt key metabolic pathways, including carbohydrate, lipid, and amino acid metabolism, and impair soil enzyme activities (e.g., urease, dehydrogenase), ultimately affecting nutrient cycling and organic matter decomposition [12]. While *Acidobacteria* and *Gemmatimonadetes* are highly susceptible, *Sphingomonadaceae*, *Actinobacteria*, and *Gammaproteobacteria* exhibit resilience. Short-term exposure (e.g., 40 days) induces structural and functional changes, but long-term impacts remain understudied. These alterations compromise essential ecosystem functions, such as nutrient cycling and soil health, with potential cascading effects on ecosystem stability and sustainability [12].

## 2. Materials and Methods

### 2.1. Soil and Root Sampling

In November 2022 and 2023, samples containing soil and roots were collected from explosive-contaminated sites at the “Depósito Conjunto de Municiones, El Corazón” in Machachi, Pichincha, Ecuador. In both years, root and soil samples were taken from three distinct areas: unpolluted soil, incineration site, and detonation site. At each site, nine sub-samples were collected randomly within a 10 m^2^ area at a depth of 10 cm using a 5 cm dia. soil corer and placed in a sterile plastic bag. Bags were transported to the laboratory in a cooler with ice packs. Soil was separated from roots and homogenized, while roots were washed with distilled water. Both root and soil samples were stored in 50 mL sterile plastic tubes and kept at 4 °C overnight before DNA extraction.

### 2.2. Amplicon Sequencing

Total DNA was extracted in triplicate from soil and root samples using the FastDNA™ SPIN Kit for Soil (MP Biomedicals, Santa Ana, CA, USA), following the manufacturer’s instructions. The extracted DNA concentration and quality were assessed using a UV/vis spectrophotometer for nano volumes (Thermo Fisher Scientific, Waltham, MA, USA) and agarose gel electrophoresis before samples were sent for high throughput amplicon sequencing [13]. The V3-V4 region of the 16S rRNA gene was amplified using universal primers 341F/805R and the ITS region with ITS3/ITS4 primers. Barcode amplification and 250 bp PE sequencing were performed by BioSequence (Quito, Ecuador) using Illumina MiSeq technology.

### 2.3. Data Analysis

All the data analyses were performed in R v 4.4.1. The obtained sequences were analyzed using the DADA2 pipeline, following the suggested protocol for Illumina amplicon data [14]. Briefly, the sequences were quality-filtered, trimmed, and clustered into amplicon sequence variants (ASVs) using the DADA2 algorithm. Taxonomic classification was performed using SILVA (16S) and UNITE (ITS) databases [15,16,17,18], employing the assign Taxonomy function in the DADA2 pipeline, with a minimum bootstrap confidence threshold of 50%. The ASVs table, the taxonomy table, and the metadata were built into a phyloseq object. Rarefaction curves were generated using the ampvis2 library [19].

The microeco library was used to build abundance heatmaps and bar plots and to calculate observed species, Shannon, Simpson, Chao1, and alpha diversity indices [20]. A Shapiro–Wilk normality test was performed separately on soil and root data for both bacteria and fungi, as it is particularly effective for small to moderate sample sizes. Based on the test results, ANOVA and Duncan tests were performed on normally distributed data to determine if alpha diversity was significantly different among treatments, and the Kruskal–Wallis test was used for non-normally distributed data. This approach ensures the appropriate application of statistical tests and minimizes the risk of Type I or Type II errors. To assess beta diversity, a Redundancy Analysis (RDA) was conducted using the mia package v 1.14 [21]. Before RDA, the ASVs table was transformed to relative abundances to account for varying library sizes across samples. The explanatory variables included in the RDA model were site (detonation, incineration, unpolluted), year (2022 or 2023), and type (root or soil), while the number of reads was also included as a continuous covariate. Bray–Curtis dissimilarity was used as the distance metric, and missing values were excluded from the analysis. The significance of the RDA results was further evaluated through PERMANOVA (Permutational Multivariate Analysis of Variance) to test the influence of each explanatory variable on the microbial community composition [22].

A differential abundance analysis of taxa was conducted using the microeco library [20]. To identify ASVs with differential abundance, the LEfSe (Linear discriminant analysis Effect Size) method was employed. The analysis focused on evaluating differences among the various sampling sites, and an alpha value of 0.05 was set for statistical significance. To ensure the robustness of the analysis, only taxa with at least 2 samples in each group were considered. Plant reads were analyzed for abundance only, as they represented just 4% of all ITS reads.

## 3. Results

### 3.1. Read Processing

ITS and 16S sequencing libraries were generated for 36 DNA samples comprising two sample types from three sites across two years, with three replicates each. Total reads per DNA sample ranged from 279 to 515,000. After filtering, ASV inference, merging, and chimera removal, the read count per sample ranged from 3 to 320,000. Rarefaction curves indicated that sufficient sequencing depth was achieved for most samples. The 16S and ITS curves approached an asymptote approximately at 15,000 and 1000 reads, respectively (Appendix A). Samples with fewer than 100 ITS reads or 800 16S reads were excluded from further analysis. These thresholds were chosen based on rarefaction curves to ensure robust diversity analyses. Raw sequence reads are available at GenBank under BioProject PRJNA1195925.

### 3.2. Microbial Communities

Bacterial communities were analyzed using 16S amplicons, while ITS was used for assessing fungal and plant communities. The taxonomic composition of bacteria, fungi, and plants varied depending on year, type of sample, and site. Bacterial composition at the class level showed that *Gammaproteobacteria* and *Bacteroidia* were dominant in most soil and root samples, except in unpolluted soils, where *Alphaproteobacteria* was the most abundant and class distribution was more even (Figure 1a). At the genus level, differences between soil and root microbiota were evident. Most notably, *Flavobacterium*, *Pseudomonas*, and *Rhizobium* were more abundant in root samples than in soil samples, while *Candidatus Udaeobacter* was common in soil samples, especially in unpolluted soil. *Streptomyces* and *Pseudoarthrobacter* were more abundant in polluted roots (Figure 2a).

Fungal taxonomic profiles based on analysis of the ITS region showed that *Dothideomycetes* and *Sordariomycetes* were the most abundant phyla in most soil and root samples, followed by *Agaromycetes,* which only had low abundance in the root samples from the detonation site. The abundance of class *Leotiomycetes* was higher in samples from unpolluted soil (Figure 1b). At the genus level, *Calophoma* was the most abundant overall. *Paraleptosphaeria* and *Fusarium* were the most abundant in root samples from both detonation and incineration sites, while *Talaromyces* was mostly found in soil samples. There were no notable differences in the abundance of any organism between contaminated and non-contaminated sites. However, *Gaeumannomyces* and *Talaromyces* appear to be more abundant in contaminated samples, while *Rhexocercosporidium* is more abundant in unpolluted samples (Figure 2b).

A subset of ITS sequences, assigned to Viridiplantae and representing 4% of the total reads, was used for the plant relative abundance analysis. Streptophyta groups, *Monocotyledoneae* and *Eudicotyledoneae* were the most abundant, but DNA from Chlorophyta classes *Chlorophyceae* and *Trebouxiophyceae* was also highly abundant in soil samples, especially in contaminated soils (Figure 1c). DNA from *Alchemilla*, *Vaccinium*, and *Hypericum* was more abundant in unpolluted samples. The plant genera *Rumex* and *Lepidium* appeared mostly at detonation and incineration sites. *Poa* was the most abundant genus in root samples from 2022 and *Trifolium* in root samples from 2023. DNA from the orchid genus *Stelis* and the green algae *Chlamydomonas* was found mostly in soil samples, regardless of contamination (Figure 2c).

### 3.3. Microbial Diversity

Observed, Shannon, Simpson, and Chao1 alpha diversity indices were calculated for both bacterial and fungal root and soil samples (Appendix A). The Shapiro–Wilk normality test indicated that most alpha diversity index data were normally distributed. No significant differences among sites were observed with indices with non-normally distributed data. In bacterial root samples, all indices showed significant differences between polluted and unpolluted sites, with unpolluted sites having the highest values of most indices, except the Simpson index for the 2023 samples, where the detonation site had the highest value. In bacterial soil samples, significant differences were found only for 2023 samples, with unpolluted sites showing the highest values across all indices. For fungi, significant differences were only observed in the Simpson index for root fungi, with higher values at unpolluted sites (Figure 3).

To assess beta diversity, we evaluated the influence of environmental variables on microbial community composition. Beta diversity RDA based on Bray–Curtis dissimilarity, followed by a PERMANOVA, was significantly influenced by multiple factors. For bacteria, the PERMANOVA model explained 40.4% of the total variance in beta diversity, with all variables significantly influencing bacterial communities (*p* < 0.05). The type of sample, root vs. soil, was the most influential factor, explaining 14.1% of the total variance (F = 6.62, *p* = 0.001), followed by site, which accounted for 11.9% of the variance (F = 2.80, *p* = 0.001). The sampling year contributed 7.7% to the variance (F = 3.64, *p* = 0.001), while the number of reads explained 4.6% (F = 2.15, *p* = 0.004). The residual variance accounted for 59.6% of the total, suggesting additional unmeasured factors affecting bacterial beta diversity (Figure 4). For fungi, the PERMANOVA model explained 39.2% of the total variance in beta diversity. The site had the most significant effect, explaining 13.1% of the variance in fungal communities (F = 2.69, *p* = 0.001). The type of sample explained 11.9% of the variance (F = 4.92, *p* = 0.001), while the sampling year contributed 7.6% (F = 3.13, *p* = 0.001). The number of reads explained 5.4% of the variance (F = 2.21, *p* = 0.011). The residual variance accounted for 60.8%, indicating that other unaccounted-for factors influenced fungal beta diversity (Figure 5).

### 3.4. Differentially Abundant Taxa

The analysis of differentially abundant taxa for both bacteria and fungi between contaminated and uncontaminated sites revealed distinct patterns in root and soil samples (Figure 6). For bacterial communities, the *Proteobacteria* phylum was dominant in both root and soil samples across all sites, being more abundant in unpolluted root and contaminated soil. However, the relative abundance of specific taxa varied. In root samples, members of the *Burkholderiales* order showed higher abundances in the detonation and unpolluted sites. The phylum *Cyanobacteria*, the families *Micrococcaceae* and *Rhizobiaceae*, and the genus *Pseudoarthrobacter* were more abundant in contaminated sites, whereas the phylum *Firmicutes* and the genus *Chitinophaga* showed differential abundance in roots from unpolluted sites. In soil samples, *Gammaproteobacteria*, *Actinobacteria*, *Sphingomonadaceae*, *and Psedomonadales* were more abundant in contaminated sites. The phylum *Acidobacteriota* and *Verrucomicrobiota* were more abundant in unpolluted soil.

For fungal communities, the class *Sordariomycetes* was more abundant in root samples from both detonation and incineration sites. On the other hand, *Leotiomycetes* and *Agaricomycetes* were more abundant in roots from unpolluted sites. Members of the family *Ceratobasidiaceae* were notably more abundant in roots from incineration sites. In soil samples, the genus *Talaromyces*, within the order *Eurotiales*, was prevalent in both detonation and incineration sites, especially in detonation sites. Fungal genera such as *Schizothecium* and *Fusarium* were more abundant in incineration sites. Meanwhile, the order *Agaricales* was found more abundantly in unpolluted soils. Interestingly, the genus *Paralepotosphaeria* was differentially abundant in roots from contaminated sites and unpolluted soils.

## 4. Discussion

This study provides significant insights into the community composition of explosive-contaminated soils, particularly highlighting the potential role of specific taxa in the natural attenuation of explosive compounds and bioremediation processes. The study reveals clear differences in the composition of bacterial and fungal communities between contaminated and unpolluted soils, with marked variations observed in both alpha and beta diversity analyses. This suggests that contamination not only affects microbial abundance but may also alter functional capabilities within the soil ecosystem.

### 4.1. Microbial Diversity and Community Composition

The dominance of *Proteobacteria* in both root and soil samples across various sites can be attributed to their versatile metabolic capabilities, which allow them to thrive in diverse environments. *Proteobacteria* are known for their role in nutrient cycling, particularly in nitrogen fixation and carbon metabolism, which are crucial for plant growth and soil health [5,6].

Significant differences were observed between contaminated and unpolluted sites, with unpolluted sites generally showing higher bacterial diversity. This observation is aligned with previous research suggesting that explosive contamination often leads to a reduction in microbial diversity [23], likely due to the selective pressure exerted by explosive compounds. In contrast, contaminated sites harbored specific taxa that may possess functional capabilities for the degradation of explosives such as TNT, RDX, and HMX. The dominance of *Gammaproteobacteria* in soil and root samples at contaminated sites supports their documented potential role in bioremediation processes [24]. Specifically, the higher abundance of *Burkholderiales* in both detonation and unpolluted sites suggests that these bacteria may play a significant role in degrading contaminants and supporting plant–microbe interactions [25]. *Burkholderiales* are known for their plant growth-promoting properties, including nitrogen fixation and production of phytohormones, which can enhance plant resilience and nutrient uptake [25]. The presence of *Cyanobacteria*, *Micrococcaceae*, *Rhizobiaceae*, and *Pseudoarthrobacter* in contaminated sites suggests their potential adaptive strategies, such as the ability to degrade complex organic pollutants and tolerate high levels of toxins [4,26]. These groups may possess bioremediative capabilities, such as the degradation of explosive compounds like TNT and RDX, making them promising candidates for environmental management and soil restoration [2,3]. Additionally, the prevalence of *Candidatus Udaeobacter* in unpolluted soils, previously identified as a bioindicator of healthy soil [27], further supports its association with soil health in our findings.

Fungal communities were similarly impacted by site conditions, with unpolluted sites showing a more even distribution of fungal classes, including *Leotiomycetes* and *Agaricomycetes*. These findings are consistent with the argument that pollution can shift fungal community composition, potentially selecting for species capable of degrading complex organic compounds [9]. The detected fungal communities at these study sites have been reported as relatively abundant fungal classes in extreme environments [28]. The observed primary distribution of *Agaricomycetes* in root samples from the detonation site may reflect their adaptive response to the unique environmental conditions created by explosive contamination. *Agaricomycetes*, which include many wood-decaying and saprotrophic fungi, are known for their ability to degrade complex organic compounds, including lignin and cellulose, which may be more abundant in disturbed or contaminated soils due to plant stress and root exudation [29,30]. The detonation site likely presents a high-stress environment with altered nutrient availability and potential accumulation of organic debris, which could favor *Agaricomycetes* due to their enzymatic capabilities to break down recalcitrant compounds [30]. Conversely, the low abundance of *Agaricomycetes* in unpolluted soils may be due to the more stable and competitive microbial community structure, where other fungal groups, such as *Leotiomycetes*, dominate. *Leotiomycetes* are often associated with less disturbed environments and are known for their roles in decomposing simpler organic matter and forming symbiotic relationships with plants [31]. In unpolluted soils, the even distribution of *Leotiomycetes* may be supported by the absence of contaminants, which allows for a more balanced microbial community with less competitive pressure from stress-tolerant fungi like *Agaricomycetes*. Additionally, unpolluted soils typically have higher organic matter quality and more stable nutrient cycling, which may favor *Leotiomycetes* due to their ecological niche in decomposing leaf litter and other plant-derived materials [31]. Thus, the distribution patterns of these fungal groups likely reflect their ecological roles and adaptive strategies in response to soil contamination and nutrient dynamics. The higher abundance of *Gaeumannomyces* in contaminated soils suggests its potential involvement in natural attenuation, although it has not been reported by previous studies. Interestingly, *Calophoma*, *Paraleptosphaeria*, and *Fusarium*, genera containing mostly plant pathogenic species, were abundant in the roots of contaminated sites. These taxa may be an indicator of contaminated environments and play a role in bioremediation, as previously described [10]. Dothideomycetes and Sordariomycetes were abundant in all samples. These fungal classes play key ecological roles in soil and root ecosystems. Dothideomycetes, versatile as saprobes, pathogens, and endophytes, contribute to organic matter decomposition and nutrient cycling, particularly in stressed environments. Sordariomycetes, known for degrading complex plant materials like lignin and cellulose, also include plant pathogens and endophytes that influence plant health and soil dynamics. In contaminated sites, both classes may aid in pollutant degradation, supporting natural attenuation and ecosystem recovery [32,33,34]. Plant taxa also differed between polluted and unpolluted sites, with greater diversity observed at unpolluted locations. In contaminated sites, *Poa* dominated in 2022, while *Trifolium* was prevalent in 2023. The ability of these genera to survive in explosive-contaminated soil highlights their potential for phytoremediation. However, it is important to consider the limitations of explosive phytoremediation, as its effectiveness has been demonstrated in laboratory settings but less so in the field [35].

### 4.2. Beta Diversity and Environmental Influence

Beta diversity analysis revealed that both bacterial and fungal communities were significantly influenced by site, sample type (root vs. soil), and sampling year. For bacterial communities, site and sample type were the most influential factors, which is consistent with previous studies that have highlighted the strong effect of contamination on microbial community structure [12]. Notably, the difference in root beta diversity between contaminated and unpolluted sites suggests a dynamic response of root-associated microbial communities to environmental changes, potentially driven by plant–microbe interactions.

The site type was the most significant component influencing beta diversity in fungal communities. Detonation and incineration sites showed distinct community compositions compared to unpolluted sites. This suggests that soil contamination alters both bacterial and fungal communities, which may affect ecosystem functions such as nutrient cycling and organic matter decomposition [9,36].

### 4.3. Implications for Natural Attenuation

The differential abundance analysis identified key bacterial and fungal taxa that may contribute to natural attenuation processes. *Pseudomonas* and *Flavobacterium* were more abundant in roots, independently of contamination, showing their capacity to thrive within roots. The enrichment of *Pseudoarthrobacter* and *Streptomyces* in root samples from contaminated sites suggests that these genera could play a role in the degradation of explosives or the detoxification of harmful by-products [36]. Additionally, the dominance of the cyanobacterial family *Micrococcaceae* and the genus *Pseudoarthrobacter* in contaminated soils indicates their potential involvement in pollutant transformation or stabilization.

Fungal taxa such as *Gaeumannomyces* and *Talaromyces*, which were more abundant in contaminated soils, may also have functional roles in breaking down explosives or facilitating the survival of plants in polluted environments. These fungi might help phytoremediation efforts by boosting plant health and resilience under stress conditions, as demonstrated in prior investigations [8]. *Rhexocercosporidium*, a genus of plant pathogenic and endophytic fungi, was more abundant in unpolluted roots and soils and in roots from polluted sites, indicating that it is not capable of surviving in polluted soils.

This study reveals distinct bacterial and fungal community compositions between polluted and unpolluted sites. While the data provide a foundation for understanding how microbial communities respond to contamination, further research is needed to elucidate the specific metabolic pathways involved in the degradation of explosives. It is important to consider that bioaugmentation strategies using native microbial consortia with the use of additives may result in faster bioremediation processes [26,37]. Additionally, functional assays and metagenomic analyses would help to identify key genes and enzymes involved in these processes, ultimately contributing to more effective bioremediation strategies. Given the global scale of soil contamination by explosives, understanding the role of microbial communities in natural attenuation is essential for the development of sustainable soil remediation technologies.

While this research provides significant insights into microbial community composition, several limitations should be acknowledged: First, the study was conducted over two years; however, microbial communities can exhibit temporal fluctuations. A longer-term study could provide a more comprehensive understanding of how microbial diversity evolves in response to contamination over time. Second, the exclusion of low-depth samples might have led to the loss of some variability, which could impact diversity analyses. However, this approach ensures reliability in the observed patterns. Third, although taxonomic profiling was performed, functional analyses of microbial communities were not conducted. Integrating metagenomic or metatranscriptomic approaches could provide insights into the metabolic capabilities of the identified taxa. Fourth, environmental variables: factors such as soil chemistry, moisture content, and temperature were not extensively analyzed. These environmental variables could significantly influence microbial community structure and could explain the data fluctuations and should be incorporated into future research. Fifth, although the ITS gene is one of the most effective markers for identifying fungal taxa at the genus level, a recent study of rhizosphere soils reported a great number that lacked genus-level classification, suggesting that the advancements in long-read sequencing technologies could enhance the accuracy of fungal taxonomic identification in future studies. Therefore, taxonomic results should be interpreted cautiously [28]. Future research could apply the ecological succession hypothesis to better understand the dynamics of this type of disturbance and recovery, which will inform better restoration and bioremediation practices at those sites.

The different observations of microbial community assembly between bacteria and fungi could be explained with experimental work applying microbial succession theory. In high mountain environments, such as the site of study, microbial communities are key players in soil formation and pioneer plant colonization and growth [38]. Bacteria are typically the first colonizers; even after disturbance, bacterial groups initiate key processes that enable ecosystem establishment and provide resources for fungi, plants, and animals to colonize later. The functional roles of these pioneer bacteria are varied; they can weather and detoxify environments to increase habitability, provide organic carbon and bioavailable nitrogen through carbon and nitrogen fixation, and allow the subsequent formation of mutualistic relationships with plant species [39]. Many bacteria are also better adapted (than are fungi) to life in barren, early-successional sediments in that some can fix nitrogen and carbon from the atmosphere—traits not possessed by any fungi [40]—suggesting that the observed microbial soil communities could be the pioneer agents that could be key players for subsequent fungi and plant colonization. The contaminated sites are subject to continuous disturbance and could be in the early stages of recovery, which may explain the absence of mycorrhizal fungi in the ITS fragment analysis.

The results of this study suggest that the microbial community structure in explosive-contaminated soils may follow a predictable successional pattern, where initial colonizers, including bacteria and fungi, establish themselves rapidly in response to contamination. The initial microbial response to explosive contamination involves a rapid shift in community structure, favoring early colonizers such as *Pseudarthrobacter* and *Rhizobium*. These taxa are well-adapted to disturbed environments due to their metabolic versatility, including the ability to degrade xenobiotic compounds and tolerate oxidative stress. *Pseudarthrobacter* species, for instance, have demonstrated the capacity to aerobically degrade 2,4,6-trinitrotoluene (TNT) in contaminated soils [41]. Similarly, *Rhizobium* species, beyond their well-documented nitrogen-fixing capabilities, may contribute to detoxification by promoting plant-microbe interactions that enhance root exudation and microbial recruitment [42]. Over time, microbial succession leads to increased diversity and stability, likely influenced by shifts in nutrient availability and competitive interactions. The observed dominance of *Poa* and *Trifolium* in contaminated sites suggests a role for plant-associated microbes in facilitating microbial recovery, potentially by enhancing rhizosphere recruitment of beneficial bacteria and fungi. Testing these hypotheses in field conditions would require long-term monitoring of microbial diversity, functional gene expression (e.g., nitrogen cycling and hydrocarbon degradation genes), and soil physicochemical parameters. Stable isotope probing (SIP) and metatranscriptomics could reveal active microbial taxa involved in successional shifts, while plant–microbiome interaction assays could elucidate the role of root exudates in microbial recruitment. Key indicators of successful microbial succession in bioremediation efforts would include a gradual increase in microbial diversity, the re-establishment of keystone taxa involved in nutrient cycling, and enhanced resilience of the community to environmental stressors.

## 5. Conclusions

This study underscores the intricate relationship between explosive contamination and microbial community dynamics. Understanding these relationships is crucial for developing effective bioremediation strategies and restoring contaminated environments. Further research addressing the outlined hypotheses and limitations will enhance our knowledge of microbial ecology in explosive-contaminated soils and its implications for environmental health.

## Figures and Tables

**Figure 1 microorganisms-13-00453-f001:**
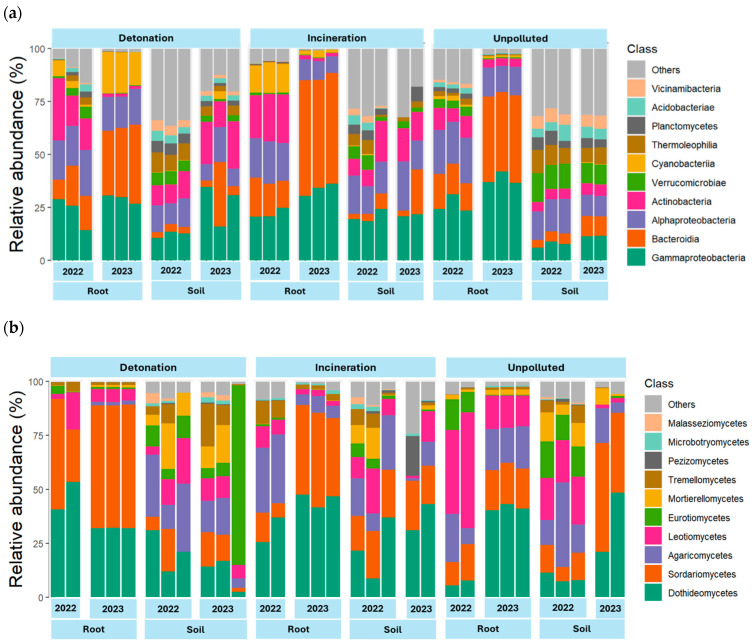
Abundance at the class taxonomic rank level for 16S and ITS barcodes: (**a**) Bacteria, (**b**) fungi, (**c**) plants.

**Figure 2 microorganisms-13-00453-f002:**
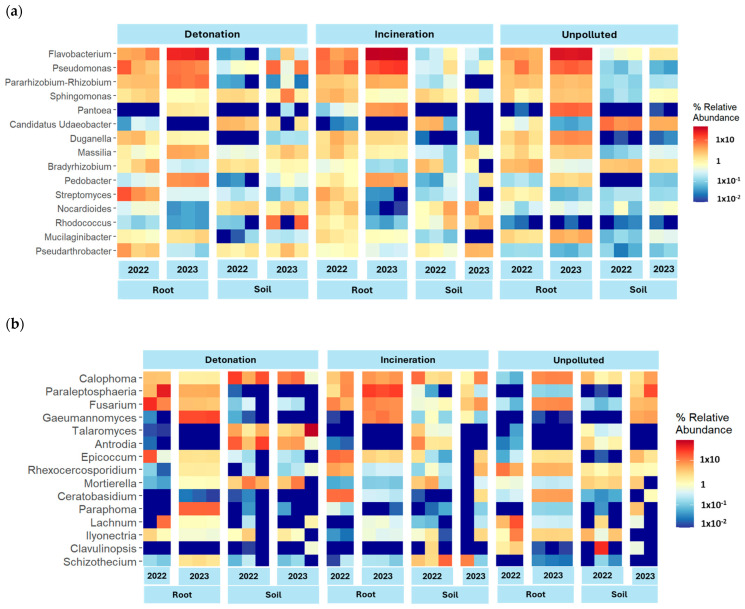
Heatmap representing the most abundant genera in all treatments: (**a**) Bacteria, (**b**) fungi, (**c**) plants.

**Figure 3 microorganisms-13-00453-f003:**
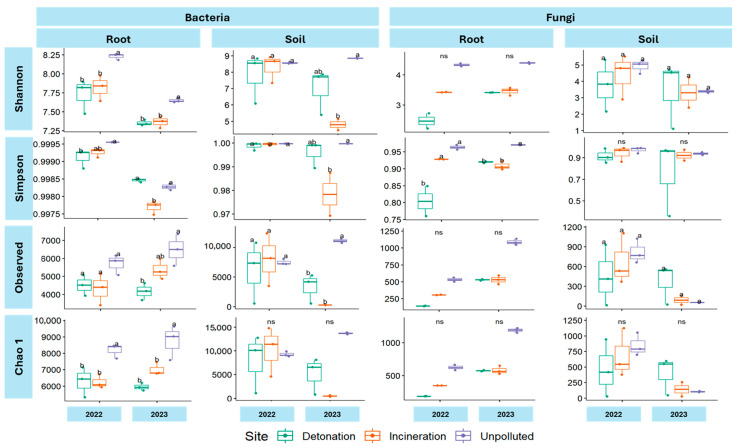
Shannon, Simpson, Observed, and Chao1 diversity indices for bacteria and fungi. A Shapiro–Wilk normality test was conducted separately on soil and root data for both bacteria and fungi. To assess whether alpha diversity differed significantly among treatments, ANOVA and Duncan tests were applied to normally distributed data, while the Kruskal–Wallis test was used for data that did not meet normality assumptions. Different letters (a, b) indicate statistically significant differences between groups (*p* < 0.05), while ’ns’ denotes non-significant differences.

**Figure 4 microorganisms-13-00453-f004:**
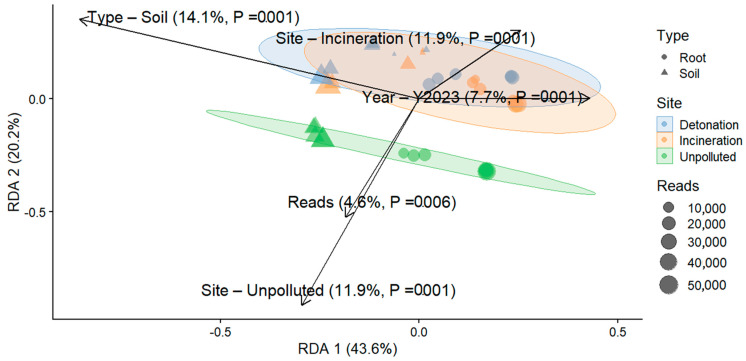
Redundancy analysis (RDA) ordination plot for bacterial communities based on Bray–Curtis dissimilarity. The vectors indicate the direction and strength of environmental variables influencing community composition, such as Site, Year, and Reads. The length of the vectors reflects the degree of correlation with the ordination axes.

**Figure 5 microorganisms-13-00453-f005:**
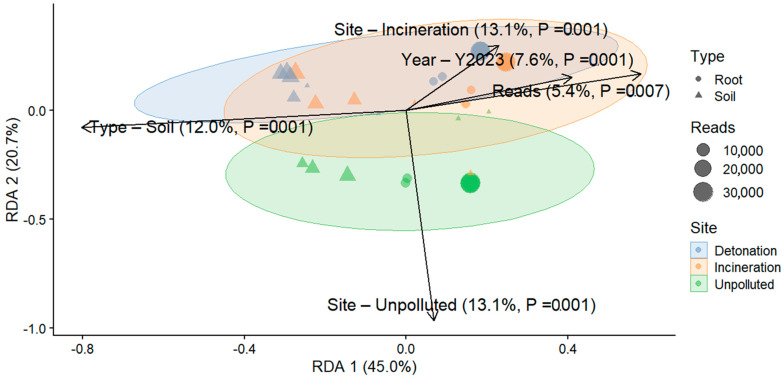
Redundancy analysis (RDA) ordination plot for fungal communities based on Bray–Curtis dissimilarity. The vectors indicate the direction and strength of environmental variables influencing community composition, such as Site, Year, and Reads. The length of the vectors reflects the degree of correlation with the ordination axes.

**Figure 6 microorganisms-13-00453-f006:**
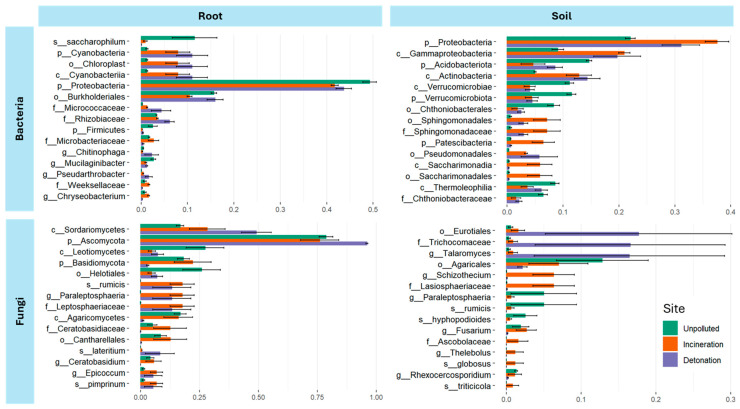
The differential abundance of bacterial and fungal taxa across root and soil samples from unpolluted, incineration, and detonation sites. Error bars represent standard deviations.

## Data Availability

Raw sequence reads are available at GenBank under BioProject PRJNA1195925. The original contributions presented in the study are included in the article/Appendix A, further inquiries can be directed to the corresponding authors.

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
