# Peer review of "Microbial Community Composition of Explosive-Contaminated Soils: A Metataxonomic Analysis"

_microorganisms, 2025, doi:10.3390/microorganisms13020453_

Round 1
Reviewer 1 Report
Comments and Suggestions for Authors
The manuscript analyzed the biodiversity of bacteria and fungi in two different soil treatments. This study is more like a research report. The author presents relevant data, but lacks in-depth analysis and innovation.
Comments on the Quality of English LanguageThe manuscript needs to be polished by native English speakers.
Author Response
Comment:
This manuscript provides an in-depth analysis of microbial communities in
soils contaminated by explosives, using metataxonomic approaches to highlight
taxa potentially involved in natural attenuation. The study is
well-structured and explores an under-researched area of microbial ecology
with implications for bioremediation and environmental health. However, there
are aspects of the study that could benefit from further clarification.
Comment:
The manuscript does not explicitly state whether the obtained sequences were
deposited in a public repository, such as GenBank, or another open-access
database. This information is crucial to ensure the reproducibility of the
study and facilitate future analyses by other researchers.
Response:
In lines 134-136 we added: Raw sequence reads are available at GenBank under BioProject PRJNA1195925.
Comment:
Sequencing depth varies significantly across samples, with some having as few
as three reads post-filtering, which could compromise the reliability of
diversity analyses. Please, clearly describe how low-depth samples were
handled and discuss the potential limitations of the findings due to this
variability.
Reponse:
In lines 133-134 we state: Samples with fewer than 100 ITS reads or 800 16S reads were excluded from further analysis. These thresholds were chosen based on rarefaction curves to ensure robust diversity analyses.
In lines 370-372 we added: The exclusion of low-depth samples might have led to the loss of some variability, which could impact diversity analyses. However, this approach ensures reliability in the observed patterns.
Comment:
The manuscript does not provide details about the confidence thresholds or
filtering criteria used for taxonomic assignment with SILVA and UNITE
databases.
Response:
In lines 96-99 now we state: Taxonomic classification was performed using SILVA (16S) and UNITE (ITS) databases [14,15,16,17], employing the assignTaxonomy function in the DADA2 pipeline, with a minimum bootstrap confidence threshold of 50%.
Comment:
The analysis of fungal communities is less detailed compared to bacterial
communities, despite their potential importance in bioremediation.
Comment:
We appreciate the reviewer’s insightful comment regarding the analysis of fungal communities in our manuscript. We agree that fungi play a critical role in bioremediation processes, and their contribution to ecosystem functioning deserves further attention. In response to the reviewer’s suggestion, In lines 284-285 now we state ¨ The detected fungal communities in this sites of study have been reported as relatively abundant fungal classes in extreme environments [28]¨.
In our study, the analysis of fungal communities was limited, as it has been reported in other studies, due to the challenges associated with fungal DNA extraction, amplification, and sequencing in comparison to bacteria, which are often more abundant and easier to analyze in environmental samples. In response to the reviwer’s suggestion, in lines 379-383 now we state ¨ Fifth, although ITS gene is one of the most effective markers for identifying fungal taxa at the genus level, a recent study of rhizosphere soils reported a great amount that lacked genus-level classification, suggesting that the advancements in long-read sequencing technologies could enhance the accuracy of fungal taxonomic identification in future studies [28].¨
We hope this addresses the reviewer’s concern, and we appreciate the opportunity to strengthen our manuscript.

Reviewer 2 Report
Comments and Suggestions for Authors
I have evaluated your manuscript and suggest it could be improved by expanding on certain sections to enhance clarity and depth. The methodology section, in particular, needs more comprehensive descriptions to help readers fully grasp your procedures. Incorporating a wider array of references would also strengthen your arguments and add richness to your study's context. Additionally, some parts of the results are unclear. Providing more clarity in these sections would improve the manuscript's effectiveness and persuasiveness.
Lines 53-55: Could you elaborate on the specific changes observed in microbial diversity and richness? What types of munition compounds were studied, and how did they specifically alter the microbial communities compared to control environments? Additionally, could you discuss the mechanisms by which these compounds influence microbial diversity? Are there any particular microbial groups that are more susceptible or resilient to these changes? It would also be helpful to know if these studies differentiated the effects based on the concentration of munition compounds and the duration of exposure. What were the long-term versus short-term impacts observed in these studies on microbial communities? Lastly, considering the environmental and potential health implications, how do these changes in microbial communities affect the broader ecosystem functions?
Lines 68-72: In detailing your DNA extraction process using the FastDNA™ SPIN Kit for Soil and assessing DNA quality with a UV/vis spectrophotometer, it would be beneficial to reference the article "https://doi.org/10.1016/j.jhazmat.2015.05.017". Citing this article could provide additional support for your methodologies, particularly in validating the extraction techniques and quality assessment measures used in your study. Integrating this reference could enhance the scientific rigor of your methodology section by demonstrating adherence to proven protocols and contributing to a more robust foundation for your high throughput amplicon sequencing results.
Lines 84-93: Could you explain why the Shapiro-Wilk normality test was chosen specifically for your data, and how it influenced the subsequent choice between ANOVA and the Kruskal-Wallis test? Additionally, when you mention alpha diversity, how were the differences among treatments quantified and interpreted? What specific trends or significant results were identified from the ANOVA and Duncan tests, and how did they contribute to your understanding of the microbial communities in different environmental treatments?
Lines 95-97: In discussing the use of PERMANOVA to evaluate the significance of your RDA results in the study of microbial community composition, it would be beneficial to include a reference to "https://doi.org/10.1016/j.marenvres.2024.106780". Citing this article can provide additional validation for the statistical methodologies employed in your analysis. Integrating this reference will enhance the credibility and robustness of your findings by linking them to a broader scientific context, thereby reinforcing the accuracy and reliability of your statistical conclusions. This citation will not only strengthen your paper but also ensure that readers understand the state-of-the-art methods used in your research.
Lines 130-139: Given your analysis of fungal taxonomic profiles from the ITS region, could you provide more insight into the ecological roles or significance of the identified fungi, particularly Dothideomycetes and Sordariomycetes, within the soil and root ecosystems? How do their functions potentially influence the environmental dynamics at both contaminated and uncontaminated sites?Additionally, the presence of Agaromycetes primarily in root samples from the detonation site suggests a specific ecological or adaptive response. Can you elaborate on why Agaromycetes might show this pattern of distribution? What environmental factors at the detonation site could be influencing their low abundance? You mention that Leotiomycetes showed higher abundance in unpolluted soil—what could be contributing to this phenomenon? Are there characteristics of unpolluted soils that particularly support Leotiomycetes, or is it a matter of competitive interactions with other fungal groups being less intense in these environments?
Lines 196-207: Your analysis highlights interesting differences in microbial community structure between contaminated and uncontaminated sites, particularly focusing on bacterial and fungal taxa. Could you elaborate on the ecological roles or potential functions of the Proteobacteria phylum that might explain their dominance in both root and soil samples across various sites? What environmental factors might influence their higher abundance in unpolluted roots compared to contaminated soil?Moreover, the presence of Burkholderiales in higher abundances at detonation and unpolluted sites within root samples is intriguing. What specific characteristics of these sites might favor Burkholderiales? And how do they potentially benefit the plant hosts in these environments? You also noted that Cyanobacteria, Micrococcaceae, Rhizobiaceae, and Pseudoarthrobacter are more prevalent in contaminated sites. What adaptive or survival strategies might these groups employ to thrive under such conditions? Additionally, could their presence indicate potential bioremediative capabilities that could be harnessed for environmental management?
Lines 242-257: Your findings reveal intriguing patterns in how site conditions influence fungal communities, with a particular focus on how contamination affects fungal diversity and species composition. Could you delve deeper into the ecological roles or adaptive strategies of Leotiomycetes and Agaricomycetes that allow them to maintain a more even distribution in unpolluted sites? What specific environmental factors at these locations support such diversity?
Lines 328-350: Your study suggests a complex interaction between microbial communities and explosive contaminants in soil, positing a successional recovery pattern in microbial assemblages. Could you elaborate on the initial changes observed in microbial community structure shortly after contamination? What specific characteristics or traits do early colonizers like Pseudarthrobacter and Rhizobium possess that allow them to thrive in such environments? Additionally, the proposed successional processes imply a transition towards greater microbial diversity and stability. What evidence supports this gradual shift, and how are nutrient availability and microbial interactions influencing these dynamics? Also, how might the presence of plant-associated taxa such as Poa and Trifolium further affect these successional processes in contaminated soils? The hypotheses surrounding plant-microbiome interactions and community stability are particularly intriguing. Could you discuss potential methods for testing these hypotheses in field conditions? What indicators would you monitor to evaluate the success of microbial succession and community stabilization in response to bioremediation efforts?Exploring these areas could provide valuable insights into the mechanisms underlying ecological restoration in contaminated sites and help optimize strategies for environmental remediation.
Author Response
Reviewer 2
I have evaluated your manuscript and suggest it could be improved by expanding on certain sections to enhance clarity and depth. The methodology section, in particular, needs more comprehensive descriptions to help readers fully grasp your procedures. Incorporating a wider array of references would also strengthen your arguments and add richness to your study's context. Additionally, some parts of the results are unclear. Providing more clarity in these sections would improve the manuscript's effectiveness and persuasiveness.
Lines 53-55: Microbial Diversity and Richness Changes
Reviewer's Question: Could you elaborate on the specific changes observed in microbial diversity and richness? What types of munition compounds were studied, and how did they specifically alter the microbial communities compared to control environments? Additionally, could you discuss the mechanisms by which these compounds influence microbial diversity? Are there any particular microbial groups that are more susceptible or resilient to these changes? It would also be helpful to know if these studies differentiated the effects based on the concentration of munition compounds and the duration of exposure. What were the long-term versus short-term impacts observed in these studies on microbial communities? Lastly, considering the environmental and potential health implications, how do these changes in microbial communities affect the broader ecosystem functions?
Response:
We have included the following paragraph in the Introduction section to address the questios from the reviewer:
Exposure to munition compounds such as TNT, RDX, and HMX significantly reduces soil microbial diversity and richness, as indicated by declines in diversity indices like the Shannon index. TNT decreases soil respiratory and enzymatic activity, favoring resilient taxa like Pseudomonas aeruginosa and Sphingomonadaceae, while reducing Acidobacteria and Gemmatimonadetes. RDX reduces soil biomass and acid phosphatase activity, increasing Actinobacteria and Bacillus, whereas HMX enhances invertase and catalase activities, promoting Gammaproteobacteria and Acidobacteria. Mixed exposures further inhibit microbial respiration and enzymatic activity, with Proteobacteria and Burkholderiaceae becoming dominant. These compounds disrupt key metabolic pathways, including carbohydrate, lipid, and amino acid metabolism, and impair soil enzyme activities (e.g., urease, dehydrogenase), ultimately affecting nutrient cycling and organic matter decomposition. While Acidobacteria and Gemmatimonadetes are highly susceptible, Sphingomonadaceae, Actinobacteria, and Gammaproteobacteria exhibit resilience. Short-term exposure (e.g., 40 days) induces structural and functional changes, but long-term impacts remain understudied. These alterations compromise essential ecosystem functions, such as nutrient cycling and soil health, with potential cascading effects on ecosystem stability and sustainability.
Lines 68-72: DNA Extraction Process
Reviewer's Question: In detailing your DNA extraction process using the FastDNA™ SPIN Kit for Soil and assessing DNA quality with a UV/vis spectrophotometer, it would be beneficial to reference the article "https://doi.org/10.1016/j.jhazmat.2015.05.017". Citing this article could provide additional support for your methodologies, particularly in validating the extraction techniques and quality assessment measures used in your study.
Response:
We have incorporated the recommended reference (DOI: 10.1016/j.jhazmat.2015.05.017) into the methodology (line 88) section to validate our DNA extraction and quality assessment protocols.
Lines 84-93: Statistical Analysis
Reviewer's Question: Could you explain why the Shapiro-Wilk normality test was chosen specifically for your data, and how it influenced the subsequent choice between ANOVA and the Kruskal-Wallis test? Additionally, when you mention alpha diversity, how were the differences among treatments quantified and interpreted? What specific trends or significant results were identified from the ANOVA and Duncan tests, and how did they contribute to your understanding of the microbial communities in different environmental treatments?
Response:
We modified the Methodology and the Results section to clarify the following: The Shapiro-Wilk test was chosen to assess the normality of our alpha diversity data because it is particularly effective for small to moderate sample sizes. Based on the test results, we used ANOVA for normally distributed data and the Kruskal-Wallis test for non-normally distributed data. This approach ensures the appropriate application of statistical tests and minimizes the risk of Type I or Type II errors.
For alpha diversity, we calculated indices such as observed species, Shannon, Simpson, and Chao1. Significant differences were observed between contaminated and unpolluted sites, with unpolluted sites generally showing higher bacterial diversity. For example, in 2023, unpolluted soil samples had significantly higher Shannon and Chao1 indices compared to contaminated sites, indicating greater species richness and evenness. These results suggest that contamination reduces microbial diversity, likely due to the selective pressure exerted by explosive compounds.
We have expanded the materials and methods, and discussion sections to include more detailed interpretations of the statistical results and their implications for understanding microbial community dynamics in different environmental conditions.
Lines 95-97: PERMANOVA and RDA
Reviewer's Question: In discussing the use of PERMANOVA to evaluate the significance of your RDA results in the study of microbial community composition, it would be beneficial to include a reference to "https://doi.org/10.1016/j.marenvres.2024.106780". Citing this article can provide additional validation for the statistical methodologies employed in your analysis.
Response:
We have added the suggested reference (DOI: 10.1016/j.marenvres.2024.106780) to support our use of PERMANOVA in evaluating the significance of RDA results (line 117).
Lines 130-139: Fungal Taxonomic Profiles
Reviewer's Question: Could you provide more insight into the ecological roles or significance of the identified fungi, particularly Dothideomycetes and Sordariomycetes, within the soil and root ecosystems? How do their functions potentially influence the environmental dynamics at both contaminated and uncontaminated sites? Additionally, the presence of Agaromycetes primarily in root samples from the detonation site suggests a specific ecological or adaptive response. Can you elaborate on why Agaromycetes might show this pattern of distribution? What environmental factors at the detonation site could be influencing their low abundance? You mention that Leotiomycetes showed higher abundance in unpolluted soil—what could be contributing to this phenomenon? Are there characteristics of unpolluted soils that particularly support Leotiomycetes, or is it a matter of competitive interactions with other fungal groups being less intense in these environments?
Response:
We have expanded the discussion with the following paragraphs:
On the ecological roles of Dothideomycetes and Sordariomycetes These fungal classes are known for their roles in decomposing complex organic matter and facilitating nutrient cycling. . Dothideomycetes, versatile as saprobes, pathogens, and endophytes, contribute to organic matter decomposition and nutrient cycling, particularly in stressed environments. Sordariomycetes, known for degrading complex plant materials like lignin and cellulose, also include plant pathogens and endophytes that influence plant health and soil dynamics. In contaminated sites, both classes may aid in pollutant degradation, supporting natural attenuation and ecosystem recovery.
The observed distribution of Agaricomycetes primarily in root samples from the detonation site may reflect their adaptive response to the unique environmental conditions created by explosive contamination. Agaricomycetes, which include many wood-decaying and saprotrophic fungi, are known for their ability to degrade complex organic compounds, including lignin and cellulose, which may be more abundant in disturbed or contaminated soils due to plant stress and root exudation. The detonation site likely presents a high-stress environment with altered nutrient availability and potential accumulation of organic debris, which could favor Agaricomycetes due to their enzymatic capabilities to break down recalcitrant compounds. Conversely, the low abundance of Agaricomycetes in unpolluted soils may be due to the more stable and competitive microbial community structure, where other fungal groups, such as Leotiomycetes, dominate. Leotiomycetes are often associated with less disturbed environments and are known for their roles in decomposing simpler organic matter and forming symbiotic relationships with plants. In unpolluted soils, the even distribution of Leotiomycetes may be supported by the absence of contaminants, which allows for a more balanced microbial community with less competitive pressure from stress-tolerant fungi like Agaricomycetes. Additionally, unpolluted soils typically have higher organic matter quality and more stable nutrient cycling, which may favor Leotiomycetes due to their ecological niche in decomposing leaf litter and other plant-derived materials. Thus, the distribution patterns of these fungal groups likely reflect their ecological roles and adaptive strategies in response to soil contamination and nutrient dynamics.
Lines 196-207: Microbial Community Structure
Reviewer's Question: Could you elaborate on the ecological roles or potential functions of the Proteobacteria phylum that might explain their dominance in both root and soil samples across various sites? What environmental factors might influence their higher abundance in unpolluted roots compared to contaminated soil? Moreover, the presence of Burkholderiales in higher abundances at detonation and unpolluted sites within root samples is intriguing. What specific characteristics of these sites might favor Burkholderiales? And how do they potentially benefit the plant hosts in these environments? You also noted that Cyanobacteria, Micrococcaceae, Rhizobiaceae, and Pseudoarthrobacter are more prevalent in contaminated sites. What adaptive or survival strategies might these groups employ to thrive under such conditions? Additionally, could their presence indicate potential bioremediative capabilities that could be harnessed for environmental management?
Response:
We have expanded the discussion with the following paragraph;
The dominance of Proteobacteria in both root and soil samples across various sites can be attributed to their versatile metabolic capabilities, which allow them to thrive in diverse environments. Proteobacteria are known for their role in nutrient cycling, particularly in nitrogen fixation and carbon metabolism, which are crucial for plant growth and soil health. Their higher abundance in unpolluted roots compared to contaminated soil may be influenced by the availability of organic carbon and the absence of toxic compounds, which can inhibit microbial activity. Burkholderiales, which are more abundant in detonation and unpolluted root samples, are known for their plant growth-promoting properties, including nitrogen fixation and production of phytohormones, which can enhance plant resilience and nutrient uptake. The presence of Cyanobacteria, Micrococcaceae, Rhizobiaceae, and Pseudoarthrobacter in contaminated sites suggests their potential adaptive strategies, such as the ability to degrade complex organic pollutants and tolerate high levels of toxins. These groups may possess bioremediative capabilities, such as the degradation of explosive compounds like TNT and RDX, making them promising candidates for environmental management and soil restoration.
Lines 242-257: Fungal Diversity and Species Composition
Reviewer's Question: Could you delve deeper into the ecological roles or adaptive strategies of Leotiomycetes and Agaricomycetes that allow them to maintain a more even distribution in unpolluted sites? What specific environmental factors at these locations support such diversity?
Response:
Questions about Agaromycetes and Leotiomycetes were addresed in the Fungal Taxonomic Profiles part of the discussion section, as described above.
Lines 328-350: Microbial Succession and Recovery
Reviewer's Question: Could you elaborate on the initial changes observed in microbial community structure shortly after contamination? What specific characteristics or traits do early colonizers like Pseudarthrobacter and Rhizobium possess that allow them to thrive in such environments? Additionally, the proposed successional processes imply a transition towards greater microbial diversity and stability. What evidence supports this gradual shift, and how are nutrient availability and microbial interactions influencing these dynamics? Also, how might the presence of plant-associated taxa such as Poa and Trifolium further affect these successional processes in contaminated soils? The hypotheses surrounding plant-microbiome interactions and community stability are particularly intriguing. Could you discuss potential methods for testing these hypotheses in field conditions? What indicators would you monitor to evaluate the success of microbial succession and community stabilization in response to bioremediation efforts?
Response:
To adrdress mthe comments we replaced the last paragraph of the discusión with the following text:
The initial microbial response to explosive contamination involves a rapid shift in community structure, favoring early colonizers such as Pseudarthrobacter and Rhizobium. These taxa are well-adapted to disturbed environments due to their metabolic versatility, including the ability to degrade xenobiotic compounds and tolerate oxidative stress. Pseudarthrobacter species, for instance, have demonstrated the capacity to aerobically degrade 2,4,6-trinitrotoluene (TNT) in contaminated soils (Kumar et al., 2021). Similarly, Rhizobium species, beyond their well-documented nitrogen-fixing capabilities, may contribute to detoxification by promoting plant-microbe interactions that enhance root exudation and microbial recruitment (Sessitsch et al., 2013). Over time, microbial succession leads to increased diversity and stability, likely influenced by shifts in nutrient availability and competitive interactions. The observed dominance of Poa and Trifolium in contaminated sites suggests a role for plant-associated microbes in facilitating microbial recovery, potentially by enhancing rhizosphere recruitment of beneficial bacteria and fungi. Testing these hypotheses in field conditions would require long-term monitoring of microbial diversity, functional gene expression (e.g., nitrogen cycling and hydrocarbon degradation genes), and soil physicochemical parameters. Stable isotope probing (SIP) and metatranscriptomics could reveal active microbial taxa involved in successional shifts, while plant-microbiome interaction assays could elucidate the role of root exudates in microbial recruitment. Key indicators of successful microbial succession in bioremediation efforts would include a gradual increase in microbial diversity, the re-establishment of keystone taxa involved in nutrient cycling, and enhanced resilience of the community to environmental stressors.

Reviewer 3 Report
Comments and Suggestions for Authors
The revised manuscript studies interesting environment - explosive-contaminated sites, the places common in many places yet not studied deeply.
Identification and description microbial communities structure of such places is very interesting and provides insights into such locations. This may help in finding microbes which may be further used in bioremediation of explosives.
For that reason I found the manuscript perfectly fitting to Microorganisms Journal and being worth to publish.
It is very well structured with a clear title, accurate keywords and concise abstract.
Introduction is so concise as it could be highlighting the problem and objective of the study.
Methodology is current and well chosen; methods are modern and typical for similar studies. All parts of this section are described in the way allowing other researchers to duplicate it. Research is supported by very good statistical and bioinformatic analysis.
Results are clear, supported with very good figures with all required data. Next is discussion explaining the results and confronting them with other studies. Literature is also well chosen with works from recent years.
All of this leads to clear and convincing conclusions opening doors for future research. In other words this work is a good start point for knowing better the potential of microbes from polluted sites to bioremediate explosives.
Author Response
Dear Reviewer,
Thank you for your positive and encouraging feedback on our manuscript. We sincerely appreciate your thoughtful comments on the relevance of our study and the potential of microbial communities in explosive-contaminated sites for bioremediation applications.
We are especially grateful for your acknowledgment that our work contributes to a deeper understanding of microbial communities in polluted environments and opens new avenues for future research in bioremediation. Your support reinforces the significance of this study.
Thank you once again for your time and insightful review. We appreciate your kind words and constructive feedback.
Round 2
Reviewer 2 Report
Comments and Suggestions for Authors
I am delighted to observe the significant improvements you have made to the manuscript. The extensive revision work has greatly enhanced the structure of the paper, as well as the clarity of the methodologies and the presentation of results.
Your efforts in refining and strengthening each section are commendable. The manuscript now provides a clear, well-articulated exploration of the topics at hand, making it a valuable contribution to the field.